# On the Relationship Between Relevance and Conflict in Online Social Link Recommendations

**Yanbang Wang**
Department of Computer Science
Cornell University
ywangdr@cs.cornell.edu

**Jon Kleinberg**
Department of Computer Science
Cornell University
kleinberg@cornell.edu

## Abstract

In an online social network, link recommendations are a way for users to discover relevant links to people they may know, thereby potentially increasing their engagement on the platform. However, the addition of links to a social network can also have an effect on the level of conflict in the network — expressed in terms of polarization and disagreement. To this date, however, we have very little understanding of how these two implications of link formation relate to each other: are the goals of high relevance and conflict reduction aligned, or are the links that users are most likely to accept fundamentally different from the ones with the greatest potential for reducing conflict? Here we provide the first analysis of this question, using the recently popular Friedkin-Johnsen model of opinion dynamics. We first present a surprising result on how link additions shift the level of opinion conflict, followed by explanation work that relates the amount of shift to structural features of the added links. We then characterize the gap in conflict reduction between the set of links achieving the largest reduction and the set of links achieving the highest relevance. The gap is measured on real-world data, based on instantiations of relevance defined by 13 link recommendation algorithms. We find that some, but not all, of the more accurate algorithms actually lead to better reduction of conflict. Our work suggests that social links recommended for increasing user engagement may not be as conflict-provoking as people might have thought.

## 1 Introduction

Recent years have seen an explosion in the usage of social media and online social networks. In 2022, an estimated 4.6 billion people worldwide used online social networks regularly [1]. Online social networks such as Facebook, LinkedIn, and Twitter are transforming the society by enabling people to exchange opinions and knowledge at an unprecedented rate and scale.

However, there are also significant concerns about the disruptive effects of social media, as people observe the growing level of polarization and disagreement in online communities. Conflicts arise over topics ranging from politics [2, 3], to entertainment [4, 5] and healthcare [6]. For our purposes in this paper, we will adopt terminology from the social sciences in distinguishing between polarization and disagreement as follows: *polarization* will refer to how much people's opinions deviate from the average, and *disagreement* will refer to how much people directly connected to each other in the social network differ in their opinions.

Reducing polarization and disagreement can be a benefit both to individual users of these platforms, who would experience less conflict and stress, and potentially also to the platforms themselves, to the extent that too much contentiousness can make the platform unattractive to users.

The root causes of increased polarization and disagreement have been the subject of much concern and debate. One popular idea is Eli Pariser's "filter bubble" theory [7]. The theory attributes

37th Conference on Neural Information Processing Systems (NeurIPS 2023).

the intensification of online conflict to the recommendation algorithms deployed by providers for maximizing user engagement. It conjectures that a person becomes more biased and cognitively blinded because recommendation algorithms keep suggesting new connections and new pieces of content associated with similar-minded people, thus solidifying the person's pre-existing bias.

The "filter bubble" theory is influential, but the empirical evidence supporting it has been limited. For example, [8] conducted experiments of hypothesis testing on the effect of personalization on consumer fragmentation, concluding that there was not enough evidence to support the "filter bubble" theory; [9] also employed a data-driven method by examining the data from a movie recommender system and checked if users may gradually get exposed to more diverse content; [10] surveyed a number of active twitter users and suggested that the offline habits of online users may actually play a more important role in creating "filter bubble". These empirical studies indicate that the magnitude of the "filter bubble" effect in social recommendation system may not be as significant as the theory's popularity implies. However, more principled understandings of the strength of "filter bubble" effect caused by social recommendations are missing from the picture.

This paper aims to explore theoretical evidence and principled characterizations of the relationship between relevance and conflict in online social link recommendations. We seek to understand **how social links recommended for maximizing user engagement (*i.e.* relevance) may shape the polarization and disagreement (*i.e.* conflict) landscape of a social network.**

We note that the "filter bubble" theory essentially counter-poses two important aspects of social link recommendations: "relevance" and "reduction of conflict". The former has been well-studied as the classic problem of link prediction [11–13]; the latter involves social accountability of link recommendations, and has received rapidly increasing attentions in recent years [14, 15]. To date, however, there has been very little attempt to study the relationship between these two aspects, in contrast to the well-established paradigms to research the relationship between relevance and novelty [16, 17], relevance and diversity [18, 19], relevance and serendipity [20, 21] in recommendations.

To theoretically analyze how opinions change in response to the addition of new links, it is necessary for us to choose a base model that specifies the basic rules of (at least approximately) how opinions propagate in social networks. There are several options for this purpose, including the Friedkin-Johnsen (FJ) model [22], the Hegselmann-Krause (HK) model [23], the voter's model [24], etc. In this work, we choose the popular FJ model which will be formally introduced in Sec.2.

There are two reasons for the FJ model to be the most suitable base model for our analytical purpose. The first is its outstanding empirical validity and practicability: according to recent surveys [25], the FJ model is the only opinion dynamics model to date on which a sustained line of human-subject experiments has confirmed the model's predictions of opinion changes [22, 26–35]. The second reason is the availability of necessary notions and definitions associated with the conflict measure: to our best knowledge, the FJ model is also the only opinion dynamics model upon which social tensions like polarization and disagreement have been rigorously defined [14], and widely accepted [15, 36–39]. Therefore, it is most meaningful to conduct in-depth theoretical analysis based on the FJ model.

**Proposed Questions.** Three questions are central to our research of the relationship between relevance and minimization of opinion conflict (polarization and disagreement) in link recommendations:

**Q1.** What are the structural features of the links that can reduce conflict most effectively? Are the features aligned with those of relevant links (*i.e.* links most likely to be accepted by users)?

**Q2.** What is the empirical degree of alignment between relevance and conflict minimization for various link recommendation algorithms executed on real-world data?

**Q3.** What are the limitations of our theoretical analysis?

**Main Results.**
For **Q1**, we first study the amount of change in opinion conflict (polarization + disagreement) caused by general link additions. We derive closed-form expressions for this, which reveal a perhaps surprising fact that purely adding social links can't increase opinion conflict. Because link additions essentially improves network connectivity, we further present a theorem that uses connectivity terms to characterize opinion conflict, leading to a conclusion aligned with the surprising fact.

To interpret the structural features of links that can reduce opinion conflict most effectively, we conduct a series of explanation work on the closed-form expressions derived for conflict change. We manage to associate the expressions' components to various types of graph distances, which are

then summarized into two criteria for finding conflict-minimizing links in social networks: (1) for a single controversial topic, conflict-minimizing links should have both end nodes as close as possible in the network and their expressed opinions on that topic as different as possible; (2) for a random distribution of controversial topics, conflict-minimizing links should have both end nodes in different "clusters" of the network while still remaining fairly well-connected with each other.

For **Q2**, we introduce a model-agnostic measure called conflict awareness to empirically evaluate a recommendation model's ability of reducing conflict. We measure conflict awareness for many link recommendation algorithms on real-world social networks. We find that, some, but not all, of the more accurate recommendation algorithms have better ability to reduce conflict more effectively.

For **Q3**, we discuss a limitations of analyzing the change of opinion conflict using the FJ model, presented in the form of a paradox. The paradox indicates that reducing conflict on social networks by suggesting friend links could actually make people more stressful and upset about their social engagement. This leaves an interesting topic for future study.

## 2 Preliminaries

### 2.1 Social Network Model

We consider a social network modeled by an undirected graph $G = (V, E)$ where $V$ is the set of nodes and $E$ is the set of links. The adjacency matrix $A = (a_{ij})_{|V| \times |V|}$ is a symmetric matrix with $a_{ij} = 1$ if link $e = (i, j) \in E$, and $a_{ij} = 0$ otherwise. In more general case $a_{ij}$ can also be a non-negative scalar representing the strength of social interaction between $i$ and $j$. The Laplacian matrix $L$ is defined as $L = D - A$ where $D = \text{diag}((\sum_{j=1}^{|V|} a_{ij})_{i=1,\dots,|V|})$ is the degree matrix. The incidence matrix $B$ is a $|V| \times |E|$ matrix whose all elements are zero except that for each link $e = (i, j) \in E$, $B_{ie} = 1$, $B_{je} = -1$ ($i, j$ interchangeable as $G$ is undirected). Given a link $e = (i, j)$, its edge (indicator) vector $b_e$ is the column vector in $B$ whose column index is $e$, $(b_e)_i = 1, (b_e)_j = -1$.

Note that modeling a social network by an undirected graph does *not* restrict the social influence between two connected people to be symmetric in both directions. In the Friedkin-Johnsen model introduced below, the amount of influence carried by a link is normalized by the social presence (node degree). Symmetry is thus broken because the two destination nodes can have different degrees.

### 2.2 Friedkin-Johnsen Opinion Model

The Friedkin-Johnsen (FJ) model is one of the most popular models for studying opinion dynamics on social networks in recent years. Its basic assumption is that each person $i$ has two opinions: an initial ("innate" [39]) opinion $s_i$ that remains fixed, and an expressed opinion $z_i$ that evolves by iteratively averaging $i$'s initial opinion and its neighbors' expressed opinions at each time step:

$$z_i^{(0)} = s_i, \qquad z_i^{(t+1)} = \frac{s_i + \sum_{j \in N_i} a_{ij} z_i^{(t)}}{1 + \sum_{j \in N_i} a_{ij}} \tag{1}$$

where $N_i$ is the neighbors of node $i$; $a_{ij}$ is the interpersonal interaction strength: in the simplest form, it takes binary (0/1) values indicating whether $i, j$ are friends with each other; more sophisticated rules for assigning continuous values also exist.

It can be proved that the expressed opinions will eventually reach equilibrium, expressed in vector form: $z^{(\infty)} = (I + L)^{-1}s$, where $z^{(\infty)}, s \in \mathbb{R}^{|V|}$ are the opinion vectors. For simplicity, this paper writes $z$ in place of $z^{(\infty)}$ as the primary focus is the equilibrium state of $z$. While expressed opinions are guaranteed to reach equilibrium, they rarely reach global consensus. Previous studies [14, 36] have extended $z$, $s$, and their interplay with $G$ into a plethora of measures to reflect various types of tension over the social network, among which the most important ones are:

- **Disagreement:** $\mathcal{D}(G, s) = \sum_{(i,j) \in E} a_{ij}(z_i - z_j)^2 = s^T(I + L)^{-1}L(I + L)^{-1}s$;
- **Polarization:** $\mathcal{P}(G, s) = \sum_{i \in V}(z_i - \bar{z})^2 = \tilde{z}^T \tilde{z} = \tilde{s}^T(I + L)^{-2}\tilde{s}$;
- **Conflict:** $\mathcal{C}(G, s) = \mathcal{D}(G, s) + \mathcal{P}(G, s) = s^T(I + L)^{-1}s$;

where the mean $\bar{z} = \frac{\sum_{i=0}^{|V|} z_i}{|V|}$, and zero-centered vector $\tilde{z} = z - \bar{z}$; likewise for $\bar{s}, \tilde{s}$. In fact, all $s$ on the right-hand-side above can be replaced by $\tilde{s}$ and the equations still hold [14]. Again, for simplicity we follow the convention to assume $s$ to be always zero-centered. The tilde accent is thus omitted.

## 2.3 Spanning Rooted Forest

A *forest* is an acyclic graph. A *tree* is a connected forest. A *rooted tree* is a tree with one marked node as its root. A *rooted forest* is a forest with one marked node in each of its component. In other words, a rooted forest is a union of disjoint rooted trees. Given a graph $G = (V, E)$, its *spanning rooted forest* is a rooted forest with node set $V$. Later in this paper, we will use various counts of spanning rooted forests to help interpret mathematical quantities arising from our analysis.

# 3 Conflict Change Caused by Link Additions

## 3.1 Link Additions Help Reduce Conflict

We start by analyzing the effect of link additions on the conflict measure of social networks. The following theorem provides closed-form expressions of both conflict change and its expected value (over random distributions of initial opinions) caused by the addition of a new link. We assume the link to have unit weight, but the result can be easily generalized to the case of continuous weights. Surprisingly, our theorem shows that both conflict change and its expected value are non-positive terms no matter which link is to be added.

**Theorem 1.** *Given initial opinions $s$ and social network $G = (V, E)$ with Laplacian matrix $L$, let $G_{+e} = (V, E \cup \{e\})$ denote the new social network. The change of conflict of expressed opinions caused by adding $e$ is given by*

$$\Delta_{+e}\mathcal{C} = \mathcal{C}(G_{+e}, s) - \mathcal{C}(G, s) = -\frac{(z_i - z_j)^2}{1 + b_e^T(I + L)^{-1}b_e} \leq 0 \tag{2}$$

*The topology term $L$ can be marginalized by considering initial opinions as independent samples from a random distribution with finite variance, i.e. $s_i \sim \mathcal{D}(0, \sigma^2)$ iid. the expected conflict change can be expressed as:*

$$\Delta_{+e}\mathbb{E}_s[\mathcal{C}] = \mathbb{E}_{s \sim \mathcal{D}(0,\sigma^2)}[\mathcal{C}(G_{+e}, s) - \mathcal{C}(G, s)] = -\frac{\sigma^2|(I + L)^{-1}b_e|_2^2}{1 + b_e^T(I + L)^{-1}b_e} \leq 0 \tag{3}$$

The marginalization of $L$ in expected conflict change allows us to focus on the effect of network structure when considering conflict change. It also reflects the fact that people may hold different initial opinions $s$ upon different controversial topics. We also computationally validate the theorem, especially to check that the conflict change caused by link additions is indeed non-positive. See Appendix C for more details.

## 3.2 Network Connectivity Helps Contract Conflict

The addition of links always improves the connectivity of a social network. Therefore, to understand the effects of link additions on conflict, it also helps to examine what network connectivity implies about conflict in general. Here is an analysis that shows having opinions propagate on a better-connected social network helps "contract" more conflict. The setup is as follows.

We create a "control group" where the effect of idea exchange over social networks gets eliminated: imagine if the same group of people that get studied are instead totally disconnected with each other, their expressed opinions $z$ would stay consistent with their initial opinions $s$ since no pressure is felt from the outside; meanwhile, their disagreement term no longer exists because the term is defined only for connected people. Therefore, the conflict of this control group $\mathcal{C}(G_0, s)$ would just be $s^T s$, polarization of initial opinions; here $G_0 = (V, \varnothing)$. Corresponding to this control group is the "treatment group" where opinions propagate on the network, with conflict rate $\mathcal{C}(G_0, s) = s^T(I + L)^{-1}s$. We now compare the two groups and have the following conflict contraction theorem:

**Theorem 2.** *Given initial opinions $s$ and social network $G = (V, E)$ with Laplacian matrix $L$, we can bound the ratio of conflict between the control and the treatment group :*

$$1 + \max_{(i,j) \in E}(d_i + d_j) \geq \frac{\mathcal{C}(G_0, s)}{\mathcal{C}(G, s)} \geq 1 + \frac{1}{2}d_{\min}h_G^2 \geq 1 \tag{4}$$

*where $d_i$, $d_j$ are degrees of nodes $i, j$; $d_{min}$ is $G$'s minimum node degree; $h_G$ is $G$'s Cheeger constant.*

Theorem 2 shows the range of the influence that a social network can have on public opinion conflict. Both bounds are expressed in relation to network connectivity measures. Remarkably, the lower

bound $\leq 1$ suggests that facilitating idea exchange almost always contracts conflict, and the range of contraction rate depends on the connectivity bottleneck $d_{\min}$ and $h_G$. In general, a larger $h_G$, meaning a better-connected network with less bottleneck, leads to a larger contraction rate — a more ideal case in terms of public benefit. See Appendix C for computational validations of the theorem.

## 4 Interpreting Features of Conflict-Minimizing Links

Last section shows that recommending new links to people helps reduce opinion conflict in general. We now proceed to investigate how different links may reduce different amount of opinion conflict. A natural question in that regard is **(Q1)** how we can characterize the *conflicting minimization* feature (*i.e.* reducing most conflict) of social links, especially in terms of their relationship with the *relevance* feature (*i.e.* the likelihood that users will accept and like the recommended link).

### 4.1 Conflict-Minimizing Links

Our characterization of conflict-minimizing links is extended from Theorem 1: $\Delta_{+e}\mathcal{C} = -\frac{(z_i - z_j)^2}{1 + b_e^T(I+L)^{-1}b_e}$ and $\Delta_{+e}\mathbb{E}_s[\mathcal{C}] = -\frac{\sigma^2|(I+L)^{-1}b_e|_2^2}{1 + b_e^T(I+L)^{-1}b_e}$. It is straightforward to see that the numerator $(z_i - z_j)^2$ is the difference between expressed opinions of node $i$ and $j$. The rest two terms, $b_e^T(I+L)^{-1}b_e$ and $\sigma^2|(I+L)^{-1}b_e|_2^2$, can be interpreted by the following two theorems.

**Theorem 3.** *Given a social network $G$ and a link $e = (i, j)$ to add, the term $b_e^T(I+L)^{-1}b_e$ measures a type of graph distance between nodes $i, j$. The distance can be interpreted by the following quantity:*

$$b_e^T(I+L)^{-1}b_e \equiv \mathcal{N}^{-1}(\mathcal{N}_{ij} + \mathcal{N}_{ji}) \tag{5}$$

- *$\mathcal{N}$ is the total number of spanning rooted forests of $G$;*
- *$\mathcal{N}_{xy}$ is the total number of spanning rooted forests of $G$, in which node $x$ is the root of the tree to which $x$ belongs, and $y$ belongs to a different tree than that $x$-rooted tree;*

Together with the interpretation of $(z_i - z_j)^2$, Theorem 3 gives the two criteria for finding conflict-minimizing links over fixed initial opinions $s$:

1. The two end nodes should be as close as possible in the network (so $1 + b_e^T(I+L)^{-1}b_e$ is small);
2. Expressed opinions at the end nodes should be as different as possible (so $(z_i - z_j)^2$ is large);

Criterion 1 is perhaps a bit surprising as one may think connecting two remote people should introduce more balanced perspectives to both of them. On the other hand, the two criteria still make much sense when viewed together: there must be something unusual about the network structure when two close friends with supposedly strong influence to each other actually hold very different opinions. The suggested conflict-minimizing link can be seen as a correction to the network structure's unusualness.

**Theorem 4.** *Given a social network $G = (V, E)$ and a link $e = (i, j)$ to add, the term $\sigma^2|(I+L)^{-1}b_e|_2^2$ also measures a type of graph distance between nodes $i$ and $j$. The distance can be interpreted by the following quantity:*

$$\sigma^2|(I+L)^{-1}b_e|_2^2 \equiv \sigma^2\mathcal{N}^{-2}\sum_{k \in V}(\mathcal{N}_{ik} - \mathcal{N}_{jk})^2 \tag{6}$$

*where $\mathcal{N}$ and $\mathcal{N}_{xy}$ follow the definitions in* **Theorem 3**.

Similar to Theorem 3, Theorem 4 also explains $\sigma^2|(I+L)^{-1}b_e|_2^2$ by a type of graph distance. However, note that there is a subtle difference between the two types of distance. The subtlety is especially important to distinguish because $\Delta_{+e}\mathbb{E}_s[\mathcal{C}]$ is the ratio between the two terms.

**Corollary 1.** *$\Delta_{+e}\mathbb{E}_s[\mathcal{C}]$ can be completely expressed by counts of different types of spanning rooted forests as defined in Theorem 3*

$$\Delta_{+e}\mathbb{E}_s[\mathcal{C}] \equiv -\sigma^2\mathcal{N}^{-1}(\mathcal{N} + \mathcal{N}_{ij} + \mathcal{N}_{ji})^{-1}\sum_{k \in V}(\mathcal{N}_{ik} - \mathcal{N}_{jk})^2 \tag{7}$$

We use Corollary 1 to give intuitive interpretations of $\Delta_{+e}\mathbb{E}_s[\mathcal{C}]$. First, notice that given a social network $G$, the number of $G$'s spanning rooted forests $\mathcal{N}$ in the denominator is a constant. Therefore, the competing terms in $\Delta_{+e}\mathbb{E}_s[\mathcal{C}]$ are $\mathcal{N}_{ij} + \mathcal{N}_{ji}$ and $\sum_{k \in V}(\mathcal{N}_{ik} - \mathcal{N}_{jk})^2$. According to Theorem 3,

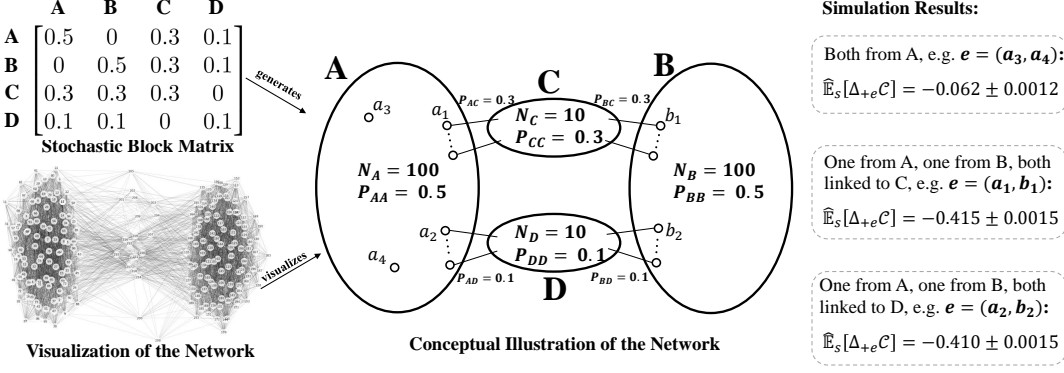

Figure 1: A barbell-like social network with cluster and bridge structures, generated from stochastic block model. Different groups of links have different structural features, producing different expected conflict change when added to network, shown in the right panel. The sample means and their 95%-intervals are reported based on repeated simulations. Note that this example gives a special network for illustrative purpose, but our interpretations of Thrm.3, 4 and Cor.1 apply to any networks.

$\mathcal{N}_{xy}$ essentially measures the distance between $x$ and $y$ by counting $x$-rooted spanning forests that separate $x$ and $y$ into different components. Therefore, $\mathcal{N}_{ij} + \mathcal{N}_{ji}$ emphasizes "local disconnectedness" between $i$ and $j$, while $\sum_{k \in V}(\mathcal{N}_{ik} - \mathcal{N}_{jk})^2$ emphasizes $i$ and $j$'s "global position gap" - "global" means that $i$ and $j$'s position is defined relative to (every node of) the entire network. The following example further illustrates their difference.

**Example 1.** *Consider a random network generated from the stochastic block model with node partitions $[N_A, N_B, N_C, N_D] = [100, 100, 10, 10]$ and block matrix shown in Figure 1 left. The diagram in Figure 1 middle illustrates the network: A, B are the two main clusters with high link density (0.5); the other two clusters, C, D, are much smaller and have lower link densities (0.1, 0.3) both internally and to A, B. Structurally speaking, C, D serve as two "bridges" between A and B.*

*Now consider adding links to three groups of previously disconnected node pairs defined as:*

- ***Group 1:*** *both end nodes in Cluster A, e.g. $(a_3, a_4)$*
- ***Group 2:*** *one end node in Cluster A, one end node in cluster B, both linked to some other node(s) in cluster C, e.g. $(a_1, b_1)$*
- ***Group 3:*** *one end node in Cluster A, one end node in cluster B, both linked to some other node(s) in cluster D, e.g. $(a_2, b_2)$*

Our simulation shows that Group 1 introduce least conflict reduction on average — in fact much less than the other two groups do. Since each pair of nodes in Group 1 are from the same densely connected cluster, it means that the numerator term $\sum_{k \in V}(\mathcal{N}_{ik} - \mathcal{N}_{jk})^2$ dominates $\Delta_{+e}\mathbb{E}_s[\mathcal{C}]$. Therefore, we may conclude that in general a link reduces more conflict if it involves two nodes that are globally distant. Comparing Group 2 and 3, we can further see that node pairs in Group 2 have stronger local connectivity than those in Group 3 ("bridge" C has higher link densities than "bridge" D). The fact that Group 2 reduces more conflict on average shows that stronger local connectivity actually positively contribute to conflict minimization.

We have now found the following two features of conflict-minimizing links, when initial opinions $s$ are sampled from a distribution with finite variance:

1. Globally, both end nodes should belong to different clusters, so that $\sigma^2 |(I + L)^{-1} b_e|_2^2$ is large;
2. Locally, both end nodes should be decently well-connected with each other, so that $1 + b_e^T(I + L)^{-1} b_e$ is small.

## 4.2 Relating Conflict Minimization to Relevance

Relevant links are links that are likely to be accepted by users. Over the past two decades, relevant links have been extensively studied as the core subject of link prediction problem in many different contexts [11–13]. We know that the relevance of social links has strong correlation with small graph distance between the two end nodes [12, 40–45]. Comparing this existing knowledge with our

characterizations of conflict-minimizing links, it is not hard to perceive that relevance and conflict minimization are not always incompatible with each other. Instead, they can have a decent degree of alignment in some cases.

# 5 Measuring the Degree of Alignment Between Relevance and Conflict Minimization on Real-World Data

Sec. 4's analysis shows that the two features of relevance and conflict minimization are *not* strictly incompatible with each other. This section discusses how we can measure the two features' degree of alignment on real-world data.

## 5.1 Definition of Conflict Awareness

We start by formulating link additions: A link addition function $f$ is defined as: $f(e; G, \beta) : (V \times V) \to [0, +\infty)$, where the function parameters are a given social network $G = (V, E)$ and a budget $\beta$ for adding links. $f$ is defined on node pairs, subject to the budget constraint $\sum_{e \in V \times V} f(e; G, \beta) = \beta$.

Among all possible link addition functions, there is a conflict-minimizing function $f^*(e; G, \beta)$, which is the function that reduces the most conflict under budget $\beta$. We use $\Delta_f \mathcal{C}$ and $\Delta_f \mathbb{E}_s[\mathcal{C}]$ to denote the conflict change and expected conflict change caused by applying $f$ over the network $G$. The two terms are related by $\Delta_f \mathbb{E}_s[\mathcal{C}] \equiv \int_s \rho(s) \, \Delta_f \mathcal{C} \, d_s$, where $\rho(s)$ is the probability density function of $s$. We further use $L_f$ to denote the Laplacian of the network formed by only the new links added by $f$.

**Definition 1.** *Given a social network $G$, initial opinions $s$, and a positivie budget $\beta$, the conflict awareness (CA) of a link addition function $f(e; G, \beta)$ is defined by the conflict reduced by applying $f$ to add links, divided by the best possible conflict reduction by applying $f^*$ to add links:*

$$\mathbf{CA}(f) = \Delta_f \mathcal{C} \,/\, \Delta_{f^*} \mathcal{C} \tag{8}$$

*where*

$$\Delta_f \mathcal{C} = s^T (I + L + L_f)^{-1} s - s^T (I + L)^{-1} s \tag{9}$$

$$\Delta_{f^*} \mathcal{C} = \min_{L_f} \ \Delta_f \mathcal{C} \tag{10}$$

$$\text{subject to} \quad L_f \in \mathcal{L} \ \textit{(Laplacian constraint)} \tag{11}$$

$$\text{Tr}(L_f) \leq 2\beta \ \textit{(budget constraint)} \tag{12}$$

$L + L_f$ is the Laplacian matrix of the network after being modified by $f$, so the definition of $\Delta_f \mathcal{C}$ is straightforward. $\Delta_{f^*} \mathcal{C}$ is defined as the objective of an optimization problem, which essentially looks for the best network with total link weights $\beta$ to be superimposed over the original network $G$.

The measure conflict awareness is useful for two reasons. First, CA essentially measures how "bad" any link recommendation algorithm is with regards to minimizing opinion conflict. By assigning $f$ a function that recommends relevant links, CA immediately becomes a quantifier that shows how much mismatch exists between $f$-suggested relevant links and conflict-minimizing links. Second, CA is a better measure than the pure conflict change $\Delta_f \mathcal{C}$. This is because CA is normalized to $[0, 1]$, which allows meaningful comparisons across different social networks. We will see that this property becomes especially helpful in practice for characterizing a link recommendation algorithm.

We can further show that CA also has a computationally desirable feature of being convex.

**Proposition 1.** *In Definition 1, $\Delta_{f^*} \mathcal{C}$ is the objective of a convex optimization problem.*

We can generalize Definition 1 to expectation of conflict awareness over a distribution of initial opinions $s$ and prove its convexity. See Appendix B.6 for details.

## 5.2 Measuring Conflict Awareness on Real-World Social Networks

### 5.2.1 Motivations and Experimental Setup

We use the measure of conflict awareness defined in Sec. 5.1 to empirically investigate two important questions on the relationship between relevance and conflict reduction in link recommendations:

- What is the conflict awareness of some of the popular off-the-shelf link recommendation algorithms? High conflict awareness (*e.g.* close to $1.0$) means that the algorithm is effective at reducing conflict, and low conflict awareness (*e.g.* less than $0.2$) means the opposite.

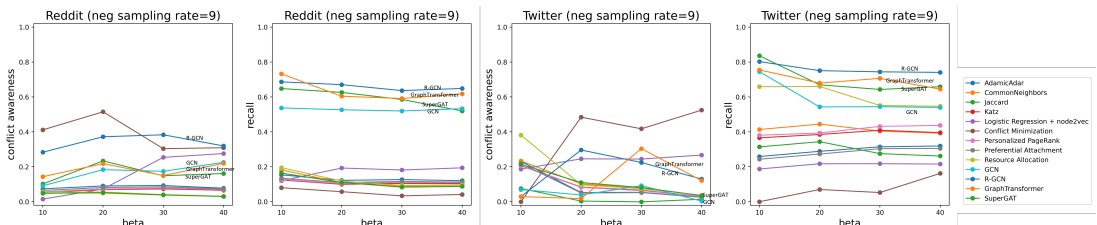

Figure 2: Measurement of conflict awareness and recall for 13 link recommendation algorithms on samples of Reddit and Twitter social networks. The x-axis is the $\eta = \frac{\#\text{negative links}}{\#\text{positive links}}$ that controls class imbalance in the test set.

- Does a more accurate link recommendation algorithm have higher conflict awareness? If we observe that for most recommendation algorithms the answer is affirmative, it means that relevance and conflict reduction are positively correlated; if we observe the opposite, it means relevance and conflict reduction are incompatible in practice.

**Datasets.** We use two real-world datasets: Reddit and Twitter, collected by [39], one of the pioneering works that conduct empirical studies on the FJ model. See Appendix C.3 for more details on how the network data and the initial opinions are generated for the two datasets.

**Baselines.** We test three groups of 13 different link recommendation methods.

- Unsupervised distance-based measures: Personalized PageRank [46], Katz Index [47], Jaccard Index [48], Adamic Adar [49], Common Neighbors [50], Preferential Attachment [50], Resource Allocation [51]. They are among the most classic methods for link recommendations, whose distance-based heuristics underpin many deep-learning-based link recommendation methods.

- Self-supervised graph learning: Logistic Regression (with Node2Vec [52] embeddings as input node features), Graph Convolutional Neural Network (GCN) [53], Relational Graph Convolutional Nerual Network (R-GCN) [54], Graph Transformer [55], SuperGAT [56]. The last three one are all GNN-based methods that achieved previous state-of-the-art on link prediction task.

- Conflict minimization solver: This is to solve the convex optimization problem defined by Eq.(10)-(12); they find optimal weights for links to be added with the sole goal of minimizing the conflict. However, it is crucial to note that this solver has total ignorance of relevance, *i.e.* it can't differentiate positive and negative links apart. Therefore, it may actually end up recommending many negative (*i.e.* invalid) links that have no effect on conflict.

**Evaluation pipeline.** For each dataset, we randomly sample $\beta = 100$ positive links from the edge set, and $\beta \cdot \eta$ negative links from all disconnected node pairs; the negative sampling rate $\eta \in [1, 10]$ is a hyperparameter. The positive links are then removed from the network to be reserved for testing, together with the negative links. It is crucial to note that in this setting (and in the real world), only a positive link can be added to the social network. If a negative link gets recommended (*i.e.* assigned positive weights), it can't be added to the network. The whole process is repeated for 10 times with different random seeds.

All links in the original network have unit weights, although the recommender is allowed to assign continuous weights to new links. As there are $\beta$ positive links, we require that the total recommended weights must sum up to $\beta$, which is enforced through linear scaling of $f$'s output. See Appendix C.5.

**Evaluation metrics.** Besides conflict awareness, we also measure the recall and precision@10 of each link recommendation method as proxies for "relevance".

**Reproducibility:** Our code and data can be downloaded from here. Other configurations of the numerical experiment can be found in Appendix C.4

### 5.2.2  Result Analysis

Fig.2 shows the measurement results of both conflict awareness and recall rate for the 13 link recommendation algorithms on the two datasets. The precision@10 measurement results are in Appendix C. The y-axis is the negative sampling rate $\eta$. The results allow us to make the following observations.

First, we observe that the conflict awareness can vary a lot for different algorithms under different settings. For example, on Reddit's social network, R-GCN's conflict awareness can be as high as 0.95 with $\eta = 1$, while Jaccard Index's conflict awareness is mostly below 0.2.

Second, we examine the effect of $\eta$. In principle, a larger $\eta$ means it is harder to identify positive links in the test set. For both networks, we can see that the conflict awareness of most algorithms drop as $\eta$ goes up, though the four GNN-based methods seem to be slightly less affected. Similar trends can be observed from the recall plots (b) and (d), where GNN-based algorithms stay quite robust as the task gets increasingly harder. These observations suggests that the ability to suggest "relevant" links can be crucial for maintaining good conflict awareness, especially when the recommendation task is hard.

Third, we have interesting observations on the "Conflict Minimization" algorithm: on both networks, this algorithm consistently produces the worst recall among all algorithms — barely usable from the perspective of relevance. However, although there are not many relevant links in its recommendation, the algorithm still has the best conflict awareness most of the time. among the algorithm's recommended links, the few ones that are actually relevant (positive) are highly effective to reduce conflict. In that sense, we can still clearly see the there exists a certain amount of misalignment between "relevance" and "conflict reduction".

## 6  Limitation: the Paradox of Conflict and Happiness

We found a limitation in the long line of existing works analyzing opinion conflict based on the FJ model, which our work also inherits from. The conflict measure we've discussed so far may *not* reflect user's happiness in their social engagement. Originally proposed in the seminal work [57], the *happiness* measure (or equivalently the *unhappiness*) quantifies the amount of mental pressure felt by people in their social engagement, as the sum of two terms: the amount of disagreement with friends (*i.e.* our disagreement term), and the amount of opinion shift between initial opinions and expressed opinions, *i.e.* internal conflict, defined as

$$\mathcal{I}(G,s) = \sum_{i \in V} (z_i - s_i)^2 = s^T \big( (I + L)^{-1} - I \big)^2 s$$

We use $\mathcal{U}(G,s)$ to denote the unhappiness of people over a social network $G$ and initial opinions $s$. Fig. 3 illustrates the relationship of important concepts in the FJ model we've discussed so far.

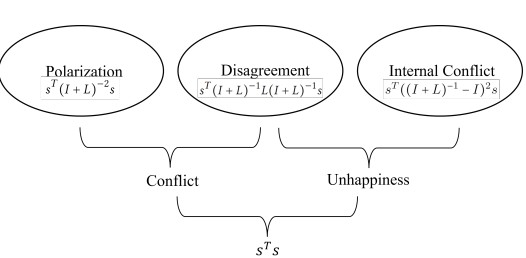

Fig.3 shows that the common term, disagreement, is shared by both measures of polarization and disagreement. Therefore, it seems likely that when one measure changes (say conflict drops due to the addition of a new link), the other measure should change in the same direction. Surprisingly however, this is not true. In fact, the following theorem shows the two measures to **always** change in opposite directions when network structure changes!

Figure 3: Relationship of the important concepts in the Friedkin-Johnsen opinion model.

**Theorem 5.** *Given initial opinions $s$ and social network $G = (V, E)$, we have the following conservation law about conflict and unhappiness (notice the RHS is independent of the graph structure):*

$$\mathcal{C}(G,s) + \mathcal{U}(G,s) = s^T s \tag{13}$$

Theorem 5 reveals a paradox: reducing conflict by modifying the structure of a social network always comes at the expense of more unhappiness of people. To resolve this paradox goes beyond this paper's scope, but would be a very interesting topic to study in the future.

## 7  Conclusion

In this work, we analyzed the relationship between relevance and opinion conflict in online social link recommendations. We present multiple pieces of evidence challenging the view that the two objectives are totally incompatible in link recommendations. For future work, it would be extremely interesting to study recommendation algorithms that can combine the two features. We also conjecture that rigorous bounds can be derived for the conflict awareness of some classical link recommendation methods such as Peronalized PageRank and Katz Index.

## Acknowledgement

The authors thank Eva Tardos and Sigal Oren for their helpful feedback on this work. This work is supported in part by a Simons Investigator Award, a Vannevar Bush Faculty Fellowship, MURI grant W911NF-19-0217, AFOSR grant FA9550-19-1-0183, ARO grant W911NF19-1-0057, a Simons Collaboration grant, and a grant from the MacArthur Foundation.

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

# Appendix

## A  Reproducibility

Our code and data can be downloaded from `https://github.com/Abel0828/NeurIPS23-Conflict-Relevance-in-FJ`.

To mitigate this risk, we suggest social platforms take more cautious steps when deciding to reduce the exposure of one person's content feed to another, such as additional algorithmic check in background, as well as more security measures to guard against the hacking of platform's administrative authority. Researchers are also encouraged to study network structures that are more robust to attacks of such kind, as well as defense measures to be taken when such attacks actually happen.

## B  Proofs

### B.1  Proof of Theorem 1

*Proof.* Let $L_{+e}$ denote the Laplacian matrix of the new social network after adding a new link $e = (i, j)$. To prove Eq.(2), we invoke the Sherman-Morrison Formula [58] for computing the inverse of rank-1 update to an invertible matrix. Notice that $G_{+e} = L + b_e b_e^T$. Therefore,

$$
\begin{aligned}
\mathcal{C}(G_{+e}, s) - \mathcal{C}(G, s) &= s^T((I + G_{+e})^{-1} - (I + L)^{-1})s \\
&= s^T((I + L + b_e b_e^T)^{-1} - (I + L)^{-1})s \\
&= -s^T \frac{(I + L)^{-1} b_e b_e^T (I + L)^{-1}}{1 + b_e^T (I + L)^{-1} b_e} s \\
&= -\frac{|b_e^T (I + L)^{-1} s|_2^2}{1 + b_e^T (I + L)^{-1} b_e} \\
&= -\frac{(z_i - z_j)^2}{1 + b_e^T (I + L)^{-1} b_e}.
\end{aligned}
$$

$L$ is positive semidefinite, so $(I + L)^{-1}$ is also positive semidefinite. Therefore, $1 + b_e^T (I + L)^{-1} b_e$ is positive, and so $-\frac{(z_i - z_j)^2}{1 + b_e^T (I+L)^{-1} b_e} \leq 0$.
To prove Eq.(3), we further note that

$$
\begin{aligned}
\mathbb{E}_s[\mathcal{C}(G_{+e}, s) - \mathcal{C}(G, s)] &= \mathbb{E}_s\left[-s^T \frac{(I + L)^{-1} b_e b_e^T (I + L)^{-1}}{1 + b_e^T (I + L)^{-1} b_e} s\right] \\
&= \mathbb{E}_s\left[-\frac{b_e^T (I + L)^{-1} s s^T (I + L)^{-1} b_e}{1 + b_e^T (I + L)^{-1} b_e}\right] \\
&= -\frac{b_e^T (I + L)^{-1} \mathbb{E}_s[s s^T] (I + L)^{-1} b_e}{1 + b_e^T (I + L)^{-1} b_e} \\
&= -\frac{b_e^T (I + L)^{-1} (\sigma^2 I) (I + L)^{-1} b_e}{1 + b_e^T (I + L)^{-1} b_e} \\
&= -\frac{\sigma^2 |(I + L)^{-1} b_e|_2^2}{1 + b_e^T (I + L)^{-1} b_e} \leq 0.
\end{aligned}
$$

$\square$

### B.2  Proof of Theorem 3

*Proof.* Let $M = I + L$, and let matrix $C$ be the co-factor matrix of $M$, then $(I + L)^{-1} = M^{-1} = |M|^{-1} C$. Therefore, $b_e^T (I + L)^{-1} b_e = |M|^{-1}(C_{ii} + C_{jj} - C_{ij} - C_{ji})$. $|M|$ is the determinant of matrix $M$. [59] presents a result that $|M|$ equals the total number of spanning rooted forests of $G$, and $C_{xy}$ equals the total number of spanning rooted forests of $G$, in which node $x$ and $y$ belong to the same tree rooted at $x$. The theorem is proved by substituting this previous result back into $|M|^{-1}(C_{ii} + C_{jj} - C_{ij} - C_{ji})$.  $\square$

### B.3 Proof of Theorem 4

*Proof.*

$$\sigma^2 |(I+L)^{-1} b_e|_2^2 = \sigma^2 \sum_{k \in V} (|M|^{-1} C_{ik} - |M|^{-1} C_{jk})^2$$

$$= \sigma^2 |M|^{-2} \sum_{k \in V} (C_{ik} - C_{jk})^2$$

Since $M$ is symmetric, we have $C_{ik} + N_{ik} = C_{ki} + \mathcal{N}_{ki} = C_{kk}$, $C_{jk} + \mathcal{N}_{jk} = C_{kj} + \mathcal{N}_{kj} = C_{kk}$, where $C_{kk}$ according to [59] is equal to the total number of spanning rooted forests where node $k$ is at the root of the tree to which $k$ belongs. Joining the two equations, we have $C_{ik} - C_{jk} = \mathcal{N}_{ik} - \mathcal{N}_{jk}$. Therefore,

$$\sigma^2 |(I+L)^{-1} b_e|_2^2 = \sigma^2 \mathcal{N}^{-2} \sum_{k \in V} (\mathcal{N}_{ik} - \mathcal{N}_{jk})^2$$

$\square$

### B.4 Proof of Corollary 1

*Proof.* The correctness quickly follows from substituting Equations (5, 6) into Equation (2). $\square$

### B.5 Proof of Proposition 1

*Proof.* To show that the objective is convex, we resort to the result in [60], Example 9: $X^{-1}$ is a matrix convex on the set of all nonnegative invertible Hermitian matrices. Obviously $I + L + L_f$ is nonnegative, invertible and symmetric, so it is a matrix convex. Therefore, the objective is convex. Any convex combination of Laplacians is still a Lapalacian. The trace of any convex combination of of matrices cannot exceed the trace of any members. Therefore, the feasible region is also convex. $\square$

### B.6 Expected Conflict Awareness

**Definition 2.** *Given a social network $G$ and a budget $\beta > 0$, the **conflict awareness over Expectation (CAE)** of a link addition function $f(e; G, \beta)$ is likewise defined as:*

$$\mathbf{CAE}(f) \equiv \frac{\Delta_f \mathbb{E}_s[\mathcal{C}]}{\Delta_{f^*} \mathbb{E}_s[\mathcal{C}]} \tag{14}$$

*where*

$$\Delta_f \mathbb{E}_s[\mathcal{C}] \equiv \sigma^2 [\text{Tr}((I + L + L_f)^{-1}) - \text{Tr}((I + L)^{-1})] \tag{15}$$

$$\Delta_{f^*} \mathbb{E}_s[\mathcal{C}] \equiv \min_{L_f} \quad \Delta_f \mathbb{E}_s[\mathcal{C}] \tag{16}$$

$$\text{subject to} \quad L_f \in \mathcal{L} \text{ (Laplacian constraint)} \tag{17}$$

$$\text{Tr}(L_f) \le 2\beta \text{ (budget constraint)} \tag{18}$$

**Proposition 2.** *The definition of $\Delta_f \mathbb{E}_s[\mathcal{C}]$ above is consistent with that of $\Delta_f \mathcal{C}$ in Definition 1 in the sense that they satisfy $\Delta_f \mathbb{E}_s[\mathcal{C}] \equiv \int_s \rho(s) \, \Delta_f \mathcal{C} \, ds$.*

*Proof.* Let $A$ be any square matrix of the same shape as $L$. Then $\int_s \rho(s) s^T A s \, ds = \int_s \rho(s) s^T (As) \, ds = \int_s \rho(s) \text{Tr}((As)s^T) \, ds = \int_s \rho(s) \text{Tr}(A(ss^T)) \, ds = \text{Tr}(A \int_s \rho(s)(ss^T) \, ds) = \text{Tr}(A(\sigma^2 I)) = \sigma^2 \text{Tr}(A)$. By substituting $A = (I + L + L_f)^{-1}$ and $A = (I + L)^{-1}$ into Eq. (9) respectively, the proposition is proved.

$\square$

**Proposition 3.** *In Definition 2, $\Delta_{f^*} \mathbb{E}_s[\mathcal{C}]$ is also the objective of a convex optimization problem.*

*Proof.* From the proof for Proposition 1, it suffices to only show that the $\Delta_f \mathbb{E}_s[\mathcal{C}]$ in Equation 15 is convex in $L_f$ given other variables fixed. Notice that we mentioned $\Delta_f \mathbb{E}_s[\mathcal{C}] \equiv \int_s \rho(s) \, \Delta_f \mathcal{C} \, ds$, in which $\rho(s) \ge 0$, $\Delta_f \mathcal{C}$ can be viewed as a function of $L_f$ and $s$, and is convex in $L_f$ given $s$ to be further fixed. Therefore, the integral $\Delta_f \mathbb{E}_s[\mathcal{C}]$ is also convex in $L_f$. $\square$

### B.7 Proof of Theorem 2

*Proof.* Let $0 = \lambda_1 \le \lambda_2 \le ... \le \lambda_n$ be eigenvalues of $L$ in ascending order; the eigen decomposition of $L = U\Lambda U^T$ where $\Lambda = \text{diag}([\lambda_1, ...\lambda_n])$ and $U$ is the corresponding orthornormal matrix satisfying $UU^T = I$. Notice that $(I + L)^{-1} = U(I + \Lambda)^{-1}U^T$.

$$\frac{\mathcal{C}(G_0, s)}{\mathcal{C}(G, s)} = \frac{s^T s}{s^T (I + L)^{-1} s} = \frac{s^T U U^T s}{s^T U (I + \Lambda)^{-1} U^T s}$$

let $s' = U^T s$, and further notice that since we have assumed $s$ to be zero-centered (see Section2), $s'_1 = 1^T s = 0$. We can further rewrite:

$$\frac{\mathcal{C}(G_0, s)}{\mathcal{C}(G, s)} = \frac{s'^T s'}{s'^T (I + \Lambda)^{-1} s'} = \frac{\sum_{i=1}^n s'^2_i}{\sum_{i=1}^n (1 + \lambda_i)^{-1} s'^2_i} = \frac{\sum_{i=2}^n s'^2_i}{\sum_{i=2}^n (1 + \lambda_i)^{-1} s'^2_i}$$

It is not hard to see that

$$1 + \lambda_n \ge \frac{\mathcal{C}(G_0, s)}{\mathcal{C}(G, s)} \ge 1 + \lambda_2$$

For the upper bound, [61] shows that $\lambda_n \le \max_{(i,j)\in E}(d_i + d_j)$; for the lower bound, we know from Lemma A.1 of [38] that $\lambda_2 \ge \frac{1}{2} d_{\min} h_G^2$, where $d_{min}$ is the minimum node degree in $G$; $h_G$ is the Cheeger constant of $G$. Substituting these back into expression above, we have $1 + \max_{(i,j)\in E}(d_i + d_j) \ge \frac{\mathcal{C}(G_0,s)}{\mathcal{C}(G,s)} \ge 1 + \frac{1}{2} d_{\min} h_G^2 \ge 1$. $\qquad\square$

### B.8 Proof of Theorem 5

*Proof.* Notice that $\mathcal{C} = s^T (I + L)^{-1} s$, and $\mathcal{U} = \mathcal{P} + \mathcal{I} = s^T (I + L)^{-2} s + s^T (I - (I + L)^{-1})^2 s = s^T (I - (I + L)^{-1}) s$. Therefore, $\mathcal{C} + \mathcal{U} = s^T s$ which is a constant. $\qquad\square$

## C Experiments

### C.1 Verifying the Direction of Conflict Change (Theorem 1)

We computationally verify that opinion conflict always gets reduced when a new link is added to the network. We use six datasets, including three synthetic networks and three real-world social networks. The synthetic networks are, a Erdős–Rényi Graph ($n = 100, p = 0.5$), a path graph ($n = 100$), a 10 by 10 2D-grid graph. The real-world networks are, the Karate club social network, Reddit, and Twitter (as introduced in Sec.C.3). For each network, we compute the amount of conflict change caused by adding a link between every pair of disconnected node in the graph, with each link replaced one at a time. Figure 4 shows the distributions of the amounts of conflict conflict for all the six datasets. We can see that they are all on the negative side of the axis. This result validates the negative sign in Theorem 1 and demonstrates its broad applicability.

### C.2 Verifying Conflict Contraction (Theorem 2)

We start with an empty graph with $N$ nodes. In each iteration, one edge is randomly added between two disconnected nodes; we then compute the the lower bound, the upper bound, and the conflict contraction rate as given in Theorem 2. The iterations stop when no pair of nodes are left disconnected (*i.e.* the graph is complete). We choose $N = 20$ in this experiment as computing the Cheeger constant term is NP-hard.

Figure 5 plots the lower bounds, the lower bounds, the upper bounds, and the conflict contraction rates, with respect to the increasing numbers of edges in the graph. We can see that the conflict contraction rates are indeed lying in between the two bounds. The gap exists because we cannot exhaust all the possible graphs on 20 nodes. Nevertheless, this experiment provides a good piece of evidence that Theorem 2 is correct.

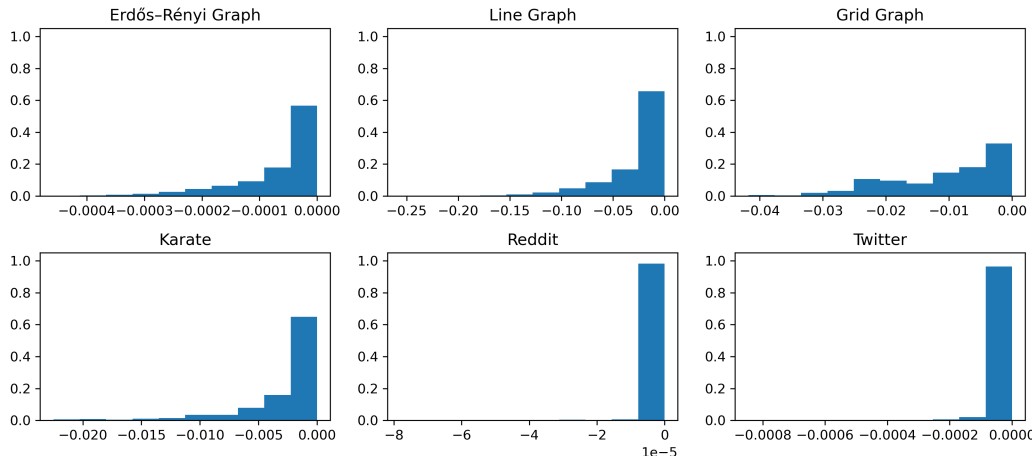

Figure 4: Computational validation for Theorem 1.

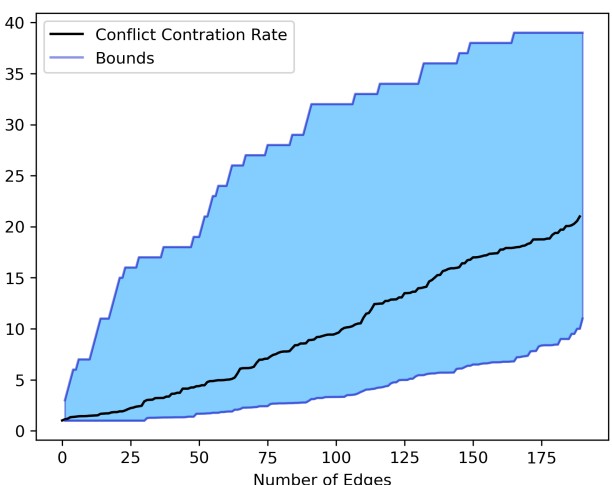

Figure 5: Computational validation for Theorem 2.

## C.3 Dataset and Preprocessing

**Twitter.** The dataset is extracted from a number of tweets relevant to the Delhi Assembly elections 2013. In the preprocessing, only the largest strongest-connected component (SCC) gets retained, which contains 548 users and 3638 undirected edges; each edge represents a pair of follower and followee. The initial opinions ($s$) were mapped by a sentiment analysis tool designed for Twitter [62], based on each user's first-hour tweets in the record window.

**Reddit.** The dataset is extracted from the subreddit of "Politics" between 07/2013 and 12/2013. Similar to Twitter, only the largest SCC is retained, containing 556 users and 8969 edges. An edge exists between two users if both of them posted in the same subreddit other than "Politics" during the aforementioned time period. The initial opinions were mapped using the standard linguistic analytics tool LIWC [63].

## C.4 Configurations

We run all experiments on Intel Xeon Gold 6254 CPU@3.15GHz with 1.6TB Memory. 7 out of the 13 methods being evaluated for the conflict awareness metric involve hyperparameter settings. They can also be found in our code: we use alpha=0.5 for Katz; alpha=0.85 for Personalized PageRank; for node2vec, we use 64 as node embedding dimensions, context window $= 10, p = 2, q = 0.5$; for GCN

and R-GCN, we use two layers with 32 as hidden dimensions, followed by a 2-layer MLP with 16 hidden units; for SuperGAT, we use two layers with 64 as hidden dimensions; for Graph Transformer, we use two layers with 32 as hidden dimensions, and attention head number of 4.

## C.5 Linear Scaling of the Output

To make sure that the weights of all recommended links sum up to $\beta$, we linearly scale each link recommendation algorithm's output by a normalizing constant: Notice that each link recommendation algorithms is essentially a scoring function on the links. For a model $h$, its output weight $w_h(e)$ of each recommended link $e$ follows the normalized form $w_h(e) = \beta \frac{s_h(e)}{\sum_e s_h(e)}$, where $s_h(e)$ is the original score that model $h$ assigns to link $e$.

## C.6 Precision@10

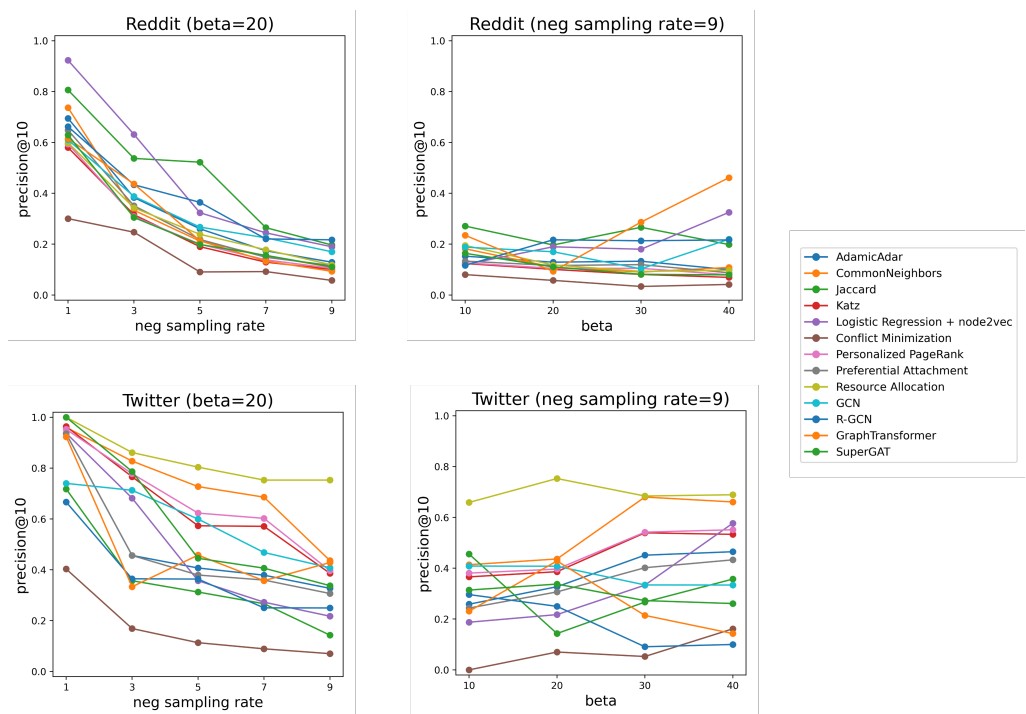

Figure 7: Precision@10 of 13 link recommendation algorithms on samples of Reddit (upper) and Twitter (lower) social network. These plots supplement the recall measurement in Fig. 2 as another proxy for "relevance".

# D Broader Impacts

Our analysis reveals the types of social links that, when added to the social network, can most effectively reduce polarization and disagreement. While this result itself is for a good cause, a potential risk exists when one interprets it into the opposite direction: we now know certain types of links that, when removed from the social network, can most effectively **increase** polarization and disagreement. This could be abused by an (authoritative) adversarial to increase polarization and disagreement by diminishing social ties among certain people, or even disconnecting them.

