# OpenReview forum: "On the Relationship Between Relevance and Conflict in Online Social Link Recommendations"
_NeurIPS.cc/2023/Conference — NeurIPS 2023 poster_

### Official Review · Reviewer_4FjV · 2023-07-01

**Soundness:** 3 good
**Presentation:** 3 good
**Contribution:** 3 good
**Rating:** 6
**Confidence:** 2

**Summary:**

The paper analyzes the relationship between relevance and opinion conflict/disagreement on online social media platforms, particularly in link recommendations. Using the Friedskin-Johnsen model that models opinion dynamics, the paper shows how link additions change online conflicts and characterize the gap in conflict reduction, depending on the reduced link recommendation. The paper evaluates their approach using real-world datasets, obtained from previous work, from Twitter and Reddit.

**Strengths:**

The paper focuses on an important and timely problem related to online polarization. I liked that the paper presents all the necessary background information about their models, as well as the assumptions that they made. Also, I believe that evaluating the proposed approach on multiple social networks and multiple recommendation algorithms makes the paper and its findings more robust.


**Weaknesses:**

I have a couple of concerns with this paper. First, the formulation of the social network as an undirected graph is not very representative of what is actually happening on popular social networks like Twitter. In reality, the social network is undirected, which raises the questions on how the presented results might change when considering the fact that the graph is undirected instead of directed. I suggest to the authors to expand their discussion on this point in the paper.

Perhaps more importantly, the paper overlooks a substantial shift that happened in social media platforms. Recently, we are observing the use of AI recommendation systems that are used for recommending and delivering content to the end-users. That is, people are exposed to content mainly based on the recommendation algorithm and, to a lesser extent, to content that is coming from people that a user follows or are friends with (e.g., see “For You” algorithmic feeds in Twitter, TikTok, etc.) These recommendation algorithms are becoming an integral part of most popular social networks nowadays. Given this, I am wondering whether the proposed approach can even be applied on this use case or how the presented results change when considering that there is an algorithm that determines what each user sees. I suggest to the authors to discuss this aspect and whether the proposed approach can be leveraged to study the same problem under these considerations.


**Questions:**

1. Can the proposed approach be applied to directed graphs? If yes, what was the rationale for using undirected graphs in this work?
2. Can the proposed approach be applied when considering that there is a separate entity (i.e., recommendation algorithm) responsible for the content delivery to users?


**Limitations:**

The authors have addressed some limitations of their work. However, I strongly encourage them to have a look at the weaknesses part of this review and expand their limitations part with the two points that are described.

---

> ### Author Rebuttal · Authors · 2023-08-10
>
> We thank the reviewer for the helpful feedback!
>
> Weakness 1 & Question 1:
>
> We agree with the reviewer that there has long been a debate on whether directed graphs should be used instead. We also understand that a key concern of the reviewer is that undirected graphs may fail to capture the asymmetric social influence between two connected people, e.g. an internet celebrity and her follower, on Twitter.
>
> Here, we explain why we used undirected graphs in the context of FJ model, which also accounts for the existence of a long line of work on FJ model which has made the same choice as ours: In FJ model, an undirected link does not imply symmetric social influence in both directions. For each link, the amount of influence carried forward by that link is normalized by the “social presence” (degree) of the destination node. Symmetry is thus broken because the two destination nodes can have different degrees. Here is an example to further illustrate this:
>
> Person A, who is an internet celebrity with 1M followers, is followed by person B, who is an ordinary person with 10 friends in total. Then:
>
> 1.  A directed graph denotes this relationship as A --> B, meaning that the social influence primarily goes one-way from A to B.
>
> 2.  An undirected graph in FJ model denotes this relationship as A --- B. However, according to Equation (1), the influence of B’s opinion on A’s has weight 1/11, and the influence of B’s on A’s has weight 1/1000001. Therefore, FJ model captures that A’s influence on B is 90909 times stronger than the B’s influence on A. In other words, the social influence still primarily goes from A to B.
>
> We would be happy to follow the reviewer’s advice to expand Part II of our “Limitations” section (Sec. 6) by the additional discussions above.
>
> Weakness 2 & Question 2:
>
> Yes, the proposed model can still be applied when considering content delivery to users. In fact, this is one of the main scenarios being covered by the FJ model, and we appreciate the reviewer for asking for clarification on this: FJ model defines a social link to be any media that carries social influence. This is a more general concept than the social links defined by a concrete relationship of “friend” or “following”. While being “a friend” with someone can certainly be modeled by a social link in the FJ model, being passively exposed to content from someone unknown (and potentially leave a comment below)because of the recommendation system can also be modeled by a social link. Tiktok’s “For You”, as mentioned by the reviewer, often recommends content from other unknown people, including internet influencers.
>
> In an even more general sense,  the spirit of FJ model can extend to the case where the recommendation system only recommends news or polls — things that are not attached to any particular individuals.  A great example is the “What’s Happening” of Twitter, which primarily recommends trending tags. In this case, the influence of one person to another is passed on in a more implicit manner, due to the “co-views”: two people that view the same content essentially becomes more connected to each other because they are influenced by the same content, have a chance to read each other’s comments, and may even start a conversation.

---

### Official Review · Reviewer_tBHa · 2023-07-02

**Soundness:** 3 good
**Presentation:** 3 good
**Contribution:** 3 good
**Rating:** 5
**Confidence:** 3

**Summary:**

This paper motivates that previous works have studied the two important aspects providing relevance and reducing conflict mainly in isolation, thus aims to explore the relationship between them. The paper uses FJ model from opinion dynamics as base model to first derive a closed form expression for quantifying changes of conflict, which is then used to analyze the conflict reduction effect caused by link addition. Based on the analysis, the paper then characterizes the conflict-reducing links and how these align with relevant links. To put the analysis in practice, a metric termed conflict-awareness is proposed to quantify the alignment between relevance and conflict minimization. The proposed metric is applied to various link recommendation algorithms on two real-world social networks. The findings show that some more accurate algorithms are indeed better at reducing conflict.

**Strengths:**

* The idea of analyzing the trade-off between relevancy and conflict in link prediction using Friedkin-Johnsen model is interesting and new to my knowledge.
* Design choices of the presented analysis are properly justified.
* The paper is easy to follow.

**Weaknesses:**

* The main claim in abstract "To this date, however, we have very little understanding of how these two implications of link formation relate to each other..." is not fully justified. A 2021 PNAS paper titled "Link recommendation algorithms and dynamics of polarization in online social networks" also tackle the issue of polarization and conflict in online social networks, specifically exploring the role that link recommendation algorithms play. Without establish the link to this 2021 paper as well as the papers mentioned therein, the main claim appeals unconvincing.

**Questions:**

* How would the paper been positioned given the PNAS 2021 paper?
* About the limitation of the proposed approach, I am wondering whether any observation can be made by dropping the equilibrium assumption of the Friedkin-Johnsen model? It would be interesting to analyze the conflict in the network form the time point of adding a link until the equilibrium is reached.
* F&J model assumes the aggregated opinion would be assimilated internally, but in some cases when introducing two opinionated nodes to each other, backfire could happen. Would this type of effect complicates the analysis? or a more vague question, how would the proposed model align with the logged events / conflict proxy measures in real world data?

**Limitations:**

* The authors pointed out two constraints: a paradox that reducing conflict could potentially stress users, and the choice of using directed versus undirected graphs for studying opinion dynamics, both limitations arising from the use of the FJ model.

---

> ### Author Rebuttal · Authors · 2023-08-10
>
> We thank the reviewer for the helpful feedback!
>
> Weakness and Question 1:
>
> The PNAS 2021 paper takes a simulation-based approach to study polarization change caused by social link substitutions (i.e. “rewiring” in their term). Our paper takes a theory-driven approach to study the change of social conflict (polarization + disagreement) caused by social link additions. Our paper is related in topic to this PNAS paper, but it differs from it in several crucial ways, as follows:
>
> 1.  Subject of study. The PNAS paper focuses on polarization, while our paper studies [polarization + disagreement] as an integral proxy for social cost.
>
> 2.  Methodology. The PNAS paper draws conclusions primarily based on computational simulations over a fixed number of social network samples. Our analysis is driven by theoretical analysis, which allows us to make claims that quantify over all networks, rather than just the networks in a set of simulation runs; our results are then supported by a number of computational experiments.
>
> 3.  The belief about how a recommended link changes a social network. The PNAS paper makes the assumption that, for every new link about to be added, a (random) existing link must be deleted simultaneously, termed as “rewiring”. Our work adopts an approach that more closely matches link recommendation in practice, where clearly links can be added without another link correspondingly being deleted; a newly added link can (proportionally) diminish the weights of existing links in the neighborhood, but need not result in any deletions.
>
> Point 2 is most prominent, since it means that the PNAS paper is not making any general claims about arbitrary networks that could then in principle be compared with ours.  For example, their work cannot guide the search for links that can reduce online conflict most effectively (and our Theorem 3 and Corollary 1 address this). They also cannot explain if the observed trend of conflict change caused by link recommendations applies universally, and why. This is also the main reason for our abstract to mention that we have limited knowledge about the relationship between relevance and conflict in online social recommendation. (We are happy to modulate this statement though if the reviewer wants to)
>
>
> Question 2:
>
> We assume that the reviewer is primarily interested in theoretical insights into the pre-equilibrium states of the conflict measure. We acknowledge that this is an underexplored area with some technical challenges for theoretical analysis. In our uploaded one-page pdf, we provide a preliminary analysis that derives the pre-equilibrium conflict with regards to s, L and the number of iterations t. We note that the quadratic matrix takes the form of power iteration and maybe further simplified by being substituted by a polynomial of the eigenvalue matrix of the adjacency matrix. We conjecture that this expression will decrease if we add a new link to G, consistent with the conclusion of our Theorem 1. However, we acknowledge that at this point the full rigorous proof remains unclear to us.
>
>
> Question 3:
>
> Regarding the backfire effect:
>
> We agree that the backfire effect is very interesting to analyze. In the current FJ model, the level of backfire is quantified by the disagreement term. The consequence of backfire can be interpreted differently by different people:
>
> If one believes that backfire would have a repulsive effect on opinions, a recent 2021 paper “Opinion dynamics with backfire effect and biased assimilation” extends the FJ model to incorporate this effect. This extended FJ model shows that backfire can lead to polarization.
> If one believes that backfire leads the two disagreeing people to simply closing up the channel of conversation, then this effect is captured by bounded confidence models such as the Hegselmann-Krause model.
>
> We note that, due to great technical challenges in analyzing nonlinear dynamics, in either case of the above there is very limited follow-up work that provides principled insights into how “relevant” links added to the network may change the conflict. But we do agree that it would be extremely interesting to study in the future.
>
>
> Regarding FJ model’s empirical validity:
>
> As the references presented in lines 69-71: to this date FJ model is the only opinion dynamics model on which a sustained line of human-subject experiments has confirmed the model’s predictions of opinion changes. For example, a 2017 PNAS paper [27], shows that a variation of FJ model helps accurately predict a total of 1,288 people’s opinion dynamics, and similarly by a Science paper in 2016 [29]. Another paper [35] in 2021 Science Advances further shows that the FJ model well fits to the data of 2015 Paris negotiation on climate change and helps identify key leadership parties. In conclusion, we believe that the FJ model, despite being simple, is a highly practical model to serve as the base model for analysis.

---

> > ### Comment · Reviewer_tBHa · 2023-08-19
> >
> > Thanks for the answers, they have addressed all my concerns. I have increased my score accordingly. I have no further questions.

---

### Official Review · Reviewer_dCzc · 2023-07-04

**Soundness:** 3 good
**Presentation:** 3 good
**Contribution:** 3 good
**Rating:** 6
**Confidence:** 3

**Summary:**

With the flourishing of online social media and social networks, the effects of polarization and disagreement brought about by link recommendation methods have become increasingly pronounced. Currently, a more widespread view is the filter bubble theory. But the empirical evidence to support him is limited, and there is a lack of detailed understanding of the strength that the filter bubble effect caused by social recommendations. To deal with these issues, this paper explores the theoretical evidence and principled features of relevance and conflict in online social link recommendation.
First, the paper investigates the amount of change in opinion conflict caused by adding general links and concludes that purely adding social links does not increase opinion conflict, also it proposes two criteria for finding conflict-minimizing links in social networks; then the paper introduced Conflict awareness to evaluate the ability of a link recommendation model to reduce conflicts; finally, it discussed the limitations of the above theoretical analysis based on the FJ model.


**Strengths:**

The main strengths of this paper are the following:
1. originality
This paper explores theoretical evidence and principled features of the effects of polarization and disagreement brought about by link recommendation algorithms in social networks. It also provides the first analysis of the question of whether the most likely accepted links in social networks are fundamentally different from those that reduce conflict by using the Friedkin-johnsen model of opinion dynamics. This paper is of good originality.
2. quality
This paper has a clear research question, rich theoretical analysis, solid experiments, and analysis of the limitations of the method, which is of high quality.
3. clarity
The paper is clear, logical and well organized.
4. significance
This paper is the first to examine whether the most likely accepted links in social networks are different from those that reduce conflicts, and analyzes the relationship between relevance and opinion conflict in link recommendation, challenging the idea that the two are incompatible.


**Weaknesses:**

The theoretical analysis in this paper is rich, the experiments may be limited in space, and it is limited in presentation, but it is necessary to introduce the corresponding experimental detail and experimental settings, please increase the representation.

**Questions:**

This paper analyzes the relationship between relevance and opinion conflict in online social link recommendation and offers a different view on its utterly incompatible perspective. This paper has the following problems:

1. the title of 2.1 is more appropriately changed to Social Network Model;
2. the text in Figure 2 should be labelled for the lines, but the second small figure in Figure 2 is confusingly labelled;
3. The theoretical part of this paper is very rich, but the space of the experimental part is limited, the configuration of the experimental environment is not explained, and only the experiments on conflict awareness are given. Is it more persuasive to put Appendix D5 in the experimental part of the main text?
4.In the conclusion of p241-243, does this distance refer to a specific distance calculation or does it refer generally to a universal distance (Are other distances applicable？).
5. The formulae should be unified and standardized, and the formulae in limitations are not labelled.
6. The number of references is too many, and the number of references in the last five years is few.


**Limitations:**

This paper adequately acknowledged the limitations, with a detailed description of the limitations

---

> ### Author Rebuttal · Authors · 2023-08-10
>
> We thank the reviewer for the helpful feedback!
>
> Question 1:
>
> We will follow the reviewer’s advice to adjust the title of Sec.2.1 to “Social Network Model”
>
> Question 2:
>
> We will follow the reviewer’s advice to relabel the lines for the second figure in Figure 2.
>
> Question 3 (and Weakness):
>
> The distance term in Line 241-243 refers to the one being defined in Theorem 3, Eq. (5): conceptually speaking, it is a type of distance associated with the number of spanning rooted forests between every node in the graph and each of the two nodes of the link.
>
> We will follow the reviewer’s advice to further increase the presence of experiments in the main text:
>
> 1. We agree that Appendix D.5 can be moved to the main text.
>
> 2. We will briefly introduce the result of Appendix D.1 and D.2 in the main text.
>
> 3. Additional configurations of the experiments: We run all experiments on Intel Xeon Gold 6254 CPU@3.15GHz with 1.6TB Memory. 7 out of the 13 methods being evaluated for the conflict awareness metric involve hyperparameter settings. They can also be found in our submitted code: we use alpha=0.5 for Katz; alpha=0.85 for Personalized PageRank; for node2vec, we use 64 as node embedding dimensions, context_window=10, p=2, q=0.5; for GCN and R-GCN, we use two layers with 32 as hidden dimensions, followed by a 2-layer MLP with 16 hidden units; for SuperGAT, we use two layers with 64 as hidden dimensions; for Graph Transformer, we use two layers with 32 as hidden dimensions, and attention head number of 4.
>
> 4. We can also provide more experiments if requested by the reviewer.
>
>
> Question 4:
>
> We will follow the reviewer’s advice to further unify the formulae and label the one in the Limitations section.
>
> Question 5:
>
> We will follow the reviewer’s advice to reduce the number of references and replace some of the old ones with their recent follow-up works. For example, we will reduce some references in line 71.

---

> > ### Comment · Reviewer_dCzc · 2023-08-15
> >
> > Thanks for the response.

---

### Official Review · Reviewer_tPqS · 2023-07-08

**Soundness:** 4 excellent
**Presentation:** 4 excellent
**Contribution:** 3 good
**Rating:** 5
**Confidence:** 3

**Summary:**

Authors theoretically analyze the conflict in the social network under Friedkin-Johnsen model. Those theoretical analysis demonstrates that the addition of links reduces the conflict. In addition, the network meaning of links to minimize conflicts is presented with theoretical support. Finally, based on the conflict-aware score, authors analyze several existing link recommendation algorithms and show that the relevant link recommendation is not fully introducing conflicts.

**Strengths:**

- Thorough analysis has been made with implying the insightful explanation on the social behaviors.
- Well-defined conflict-related measurements enable the theoretical analysis as well as the empirical measurements from real-world algorithms and datasets.
- Authors propose the conflict awareness that can represents the level of conflict that a given link recommendation algorithm introduces.

**Weaknesses:**

- The claim that the addition of links decreases the conflict is somehow obvious by the opinion formation design. By the mechanism that a give node's opinion is formed by smoothing neighbors' opinions, the addition of links implies more chances of smoothing more opinions.The given model does not have the mechanism of opinions going extreme such as taking the max votes from neighbors. It would be interesting to see if the similar results hold if the aggregation of neighbor opinions is different.


**Questions:**

- In reality, relevance may affect the opinion formation. For example, in Eq (1), a_ij can be correlated with the relevance score. It would be great to see how Section 5 analysis is affected when we incorporate this.
- Figure 2 shows downward plots in general. What would be the interpretation?

**Limitations:**

Authors address that the argument is under the FJ opinion formation model. Also, authors introduce the paradox of conflict and unhappiness, which cannot be analyzed by the manuscript.

---

> ### Author Rebuttal · Authors · 2023-08-10
>
> We thank the reviewer for the helpful feedback!
>
> Question 1:
>
> We agree with the reviewer that relevance affects opinion formation: if a link has high relevance, it may likely boost opinion exchange between the two people connected by that link. The a_ij in Eq.1, defined in lines 111-113, can generalize to this case as it represents the strength of social influence. In fact, our evaluations in Sec.5 exactly use (scaled) relevance scores in recommendations to increment the values of a_ij’s. Please see Appendix D.4 for more details.
>
> Question 2:
>
> (Please also refer to Lines 342 -347 for detailed interpretations) This downward trend shows a degree of alignment between relevance and conflict detection in link recommendations, and here is why.
>
> The negative sampling rate eta on the x-axis stands for the ratio between the number of negative (fake) links and the number of positive (real) links in our evaluation set. It can range from 1 to 9 in our setting. The larger the eta, the harder it is for a recommendation algorithm to successfully recommend a link that the user will like and accept. Notice that, if a user doesn’t want to accept a recommended link (in other words, the link is not relevant), then that link will have no effect in reducing conflict. Because social links can help reduce conflict as we’ve shown, being able to suggest more relevant links with a fixed budget will generally help reduce more conflict, and thus improve Conflict Awareness. Therefore, for each recommendation algorithm, as eta goes up:
> 1.  it becomes harder to recommend links that users like to accept (i.e. relevant links),  so the recall rate goes down (the second and the fourth plot)
> 2. the conflict awareness also has a downward trend (the first and the third plot)
>
> (Noted by another reviewer, we found that the four labels in the second plot of Figure 2 are misplaced. We have revised them. Please feel free to refer to the uploaded PDF for a better view.)

---

> ### Author Response · Authors · 2023-08-11
> **Response to Weakness**
>
> Weakness:
>
> We appreciate the reviewer for sharing the understanding of FJ model.  However, we can provide an example in which FJ model does not perform trivial smoothing:
>
> Consider a social network $G_1$ with 3 nodes: $[1, 2, 3]$. It has only one edge $(1, 2)$, and initial opinions $s=[0, 0.6, 1.5]$.
>
> By definitions of conflict ($\mathcal{C}$) and polarization ($\mathcal{P}$) in lines 133, 134, $\mathcal{C}(G_1, s) = 0.390$, $\mathcal{P}(G_1, s) = 0.0467$.
>
> Adding one edge $(1,3)$ to $G_1$, it now becomes $G_2$. Now $\mathcal{C}(G_2, s) = 0.386$, $\mathcal{P}(G_2, s) = 0.0491$.
>
> So we have: $\mathcal{C}(G_1, s) > \mathcal{C}(G_1, s)$, but $\mathcal{P}(G_1, s) < \mathcal{P}(G_2, s)$! In other words, adding a link causes polarization to increase here -- though the conflict still drops (showing that our Theorem 1 still holds).
>
> Why does this happen? First, notice that the average opinion value is always (0+0.6+1.5)/3=0.7 here. In G_1, node 1 and node 3 are pulling each other towards the average, so both of them are less polarized; In G_2, when a new links forms between node 1 and node 2 (with opinion value 0.6), node 1 is pulling node 2 away from the average value of 0.7 and thus node 2 becomes more polarized. While node 1 and 3 are still less polarized, the polarization of node 2 is more dominating.
>
> **The example above shows the complexity of FJ model which does allow the opinion of node 2 to go extreme and even dominate the polarization term**. We do agree with the reviewer though that compared to the “max vote” dynamics, FJ model is less assertive in modeling the effect of going extreme as it integrates multiple consequences of social network. There could certainly be a discussion on whether we should use a more assertive model to study one particular phenomenon, or to use a more well-rounded model that covers everything a bit. Again, in this work we choose FJ model for many of its outstanding merits as detailed in lines 68-75. In our revision, we would certainly be happy to include more discussions about other possibilities of the base model, as the reviewer suggests.

---

### Author Rebuttal · Authors · 2023-08-10

Dear reviewers and meta-reviewer,

We sincerely appreciate all the reviewers for reading our work and for providing us with the great feedback! We have responded to each reviewer separately below. Our one-page pdf for rebuttal is also uploaded here. We very much look forward to engaging with all the reviewers in the discussion phase!

With all the best regards,

Authors of Paper 9519

---

### Decision · Program_Chairs · 2023-09-21

**Decision:**

Accept (poster)

**Comment:**

The paper analyzes the relationship between relevance and conflict in online social link recommendations.
The paper shows that relevance and conflict reduction are not always incompatible, but also not always aligned, and proposes a measure of conflict awareness to evaluate the trade-off between them.

The reviewers found the paper to be clear and well-structured, and the analysis to be thorough.

The reviewers also provided multiple suggestions including discussing the limitations of the proposed approach and exploring other application to directed graphs and recommendation algorithms. The authors are encouraged to follow these suggestions to further improve the paper.